# Rapid protein stability prediction using deep learning representations

**Lasse M Blaabjerg[1], Maher M Kassem[2], Lydia L Good[1], Nicolas Jonsson[1], Matteo Cagiada[1], Kristoffer E Johansson[1], Wouter Boomsma[2]\*, Amelie Stein[1]\*, Kresten Lindorff-Larsen[1]\***

[1]Linderstrøm-Lang Centre for Protein Science, Department of Biology, University of Copenhagen, Copenhagen, Denmark; [2]Center for Basic Machine Learning Research in Life Science, Department of Computer Science, University of Copenhagen, Copenhagen, Denmark

**Abstract** Predicting the thermodynamic stability of proteins is a common and widely used step in protein engineering, and when elucidating the molecular mechanisms behind evolution and disease. Here, we present RaSP, a method for making rapid and accurate predictions of changes in protein stability by leveraging deep learning representations. RaSP performs on-par with biophysics-based methods and enables saturation mutagenesis stability predictions in less than a second per residue. We use RaSP to calculate ~ 230 million stability changes for nearly all single amino acid changes in the human proteome, and examine variants observed in the human population. We find that variants that are common in the population are substantially depleted for severe destabilization, and that there are substantial differences between benign and pathogenic variants, highlighting the role of protein stability in genetic diseases. RaSP is freely available—including via a Web interface—and enables large-scale analyses of stability in experimental and predicted protein structures.

## Editor's evaluation

This is a valuable study of broad potential impact in structural biology, which will interest readers in various fields, including molecular biomedicine, molecular evolution, and protein engineering.

**\*For correspondence:**
wb@di.ku.dk (WB);
amelie.stein@bio.ku.dk (AS);
lindorff@bio.ku.dk (KL-L)

## Introduction

Protein stability, as measured by the thermodynamic free energy difference between the native and the unfolded state, is an important feature of protein structure and therefore function. Protein stability plays a critical role when trying to understand the molecular mechanisms of evolution and has been found to be an important driver of human disease (*Casadio et al., 2011*; *Martelli et al., 2016*; *Nielsen et al., 2020*). Furthermore, optimisation of protein stability is a fundamental part of protein engineering and design (*Rocklin et al., 2017*).

For a given protein, single amino acid substitutions can have substantial effects on protein stability depending on the particular substitution and the surrounding atomic environment. Assessing such variant effects can be done experimentally for example via thermal or chemical denaturation assays (*Lindorff-Larsen and Teilum, 2021*). This process, however, may be laborious and time-consuming for even a modest number of amino acid substitutions. In contrast, computational methods have been developed that can predict protein stability changes. Such methods include well-established energy-function-based methods such as for example FoldX (*Schymkowitz et al., 2005*) and Rosetta (*Kellogg et al., 2011*), or methods based on molecular dynamics simulations (*Gapsys et al., 2016*).

Machine learning models have also been developed to predict changes in protein stability and can roughly be split into two types: supervised and self-supervised models. In supervised models, experimental protein stability measurements are used as targets for model predictions (*Li et al., 2020*; *Benevenuta et al., 2021*; *Pires et al., 2014*; *Chen et al., 2020*). Supervised models are immediately appealing as they are trained directly on experimental data and are able to make predictions at the correct absolute scale. Supervised models may, however, suffer from systematic biases, which can be hard to overcome with limited experimental data. These biases relate to issues of model overfitting of the training data, biases present in the experimental data towards destabilising mutations, biases of the types of amino acid changes that have been probed experimentally, and a lack of self-consistency in model predictions (*Yang et al., 2018*; *Pucci et al., 2018*; *Usmanova et al., 2018*; *Fang, 2019*; *Stein et al., 2019*; *Caldararu et al., 2020*).

In contrast, self-supervised models can be trained without the use of experimental protein stability measurements. Typically, self-supervised models are trained to predict masked amino acid labels from structure or sequence information, thereby learning a likelihood distribution over possible amino acid types at a particular position. This learned likelihood distribution can subsequently be used directly to predict the effects of mutations on protein stability (*Lui and Tiana, 2013*; *Boomsma and Frellsen, 2017*; *Riesselman et al., 2018*; *Shin et al., 2021*; *Elnaggar et al., 2021*; *Meier et al., 2021*). Self-supervised models sidestep many of the challenges associated with training on experimental data and have shown impressive results in recent years. These models, however, often require substantial computational resources during training—and sometimes during evaluation—and generally do not make predictions at an absolute scale, which limits their practical application.

Due to the complementary strengths and weaknesses of supervised and self-supervised models, the two perspectives can often be fruitfully combined. Here we present such an approach for rapid protein stability-change predictions, combining pre-trained representations of molecular environments with supervised fine-tuning. Our method, RaSP (Rapid Stability Prediction) provides both fast

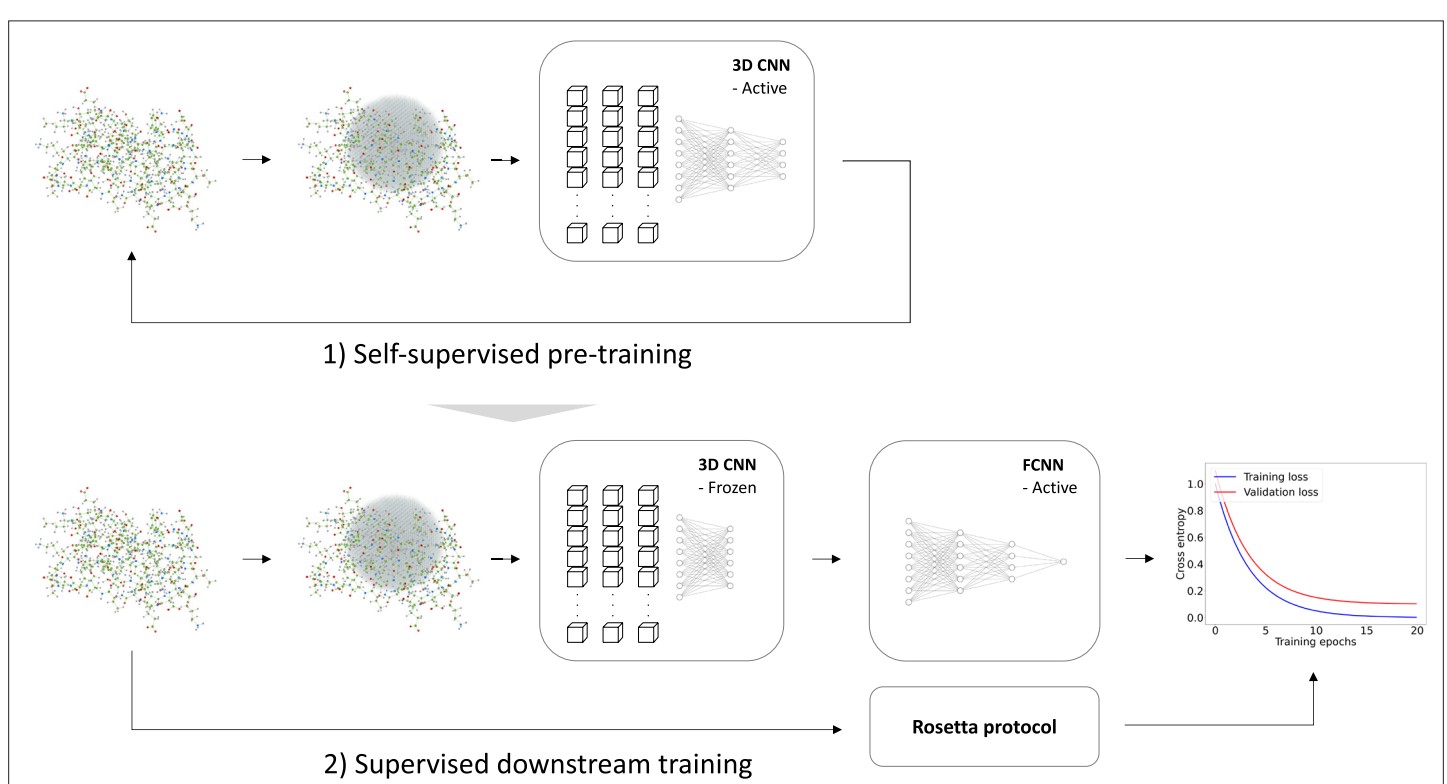

**Figure 1.** Overview of model training. We trained a self-supervised three-dimensional convolutional neural network (CNN) to learn internal representations of protein structures by predicting wild-type amino acid labels from protein structures. The representation model is trained to predict amino acid type based on the local atomic environment parameterized using a 3D sphere around the wild-type residue. Using the representations from the convolutional neural network as input, a second downstream and supervised fully connected neural network (FCNN) was trained to predict Rosetta $\Delta\Delta G$ values.

and accurate predictions of protein stability changes and thus enables large-scale, proteome-wide applications.

## Results

### Development of the RaSP model

We trained the RaSP model in two steps (*Figure 1*). First, we trained a self-supervised representation model to learn an internal representation of protein structure. We selected a 3D convolutional neural network architecture for the representation model, as this has previously been shown to yield predictions that correlate well with changes in protein stability (*Boomsma and Frellsen, 2017*). Second, we trained a supervised downstream model using the learned structure representation as input to predict protein stability changes on an absolute scale. The task of the downstream model is therefore to re-scale and refine the input from the representation model. The supervised downstream model was trained on a set of calculated protein stability changes to enable the generation of a larger and more powerful training data set while minimizing the effects from experimental data biases. We chose to estimate protein variant stability values using the Rosetta 'cartesian_ddg' protocol, which has shown to be a reliable and well-established predictor of stability (*Park et al., 2016*; *Frenz et al., 2020*), and as shown below the model generalizes to a wide range of situations. We expect that similar results could have been obtained using other variant stability prediction methods such as FoldX.

Building on earlier work (*Boomsma and Frellsen, 2017*), we first trained a self-supervised 3D convolutional neural network on a large, homology-reduced set of high-resolution structures. This model was trained to predict the wild type amino acid labels of a protein given its local atomic

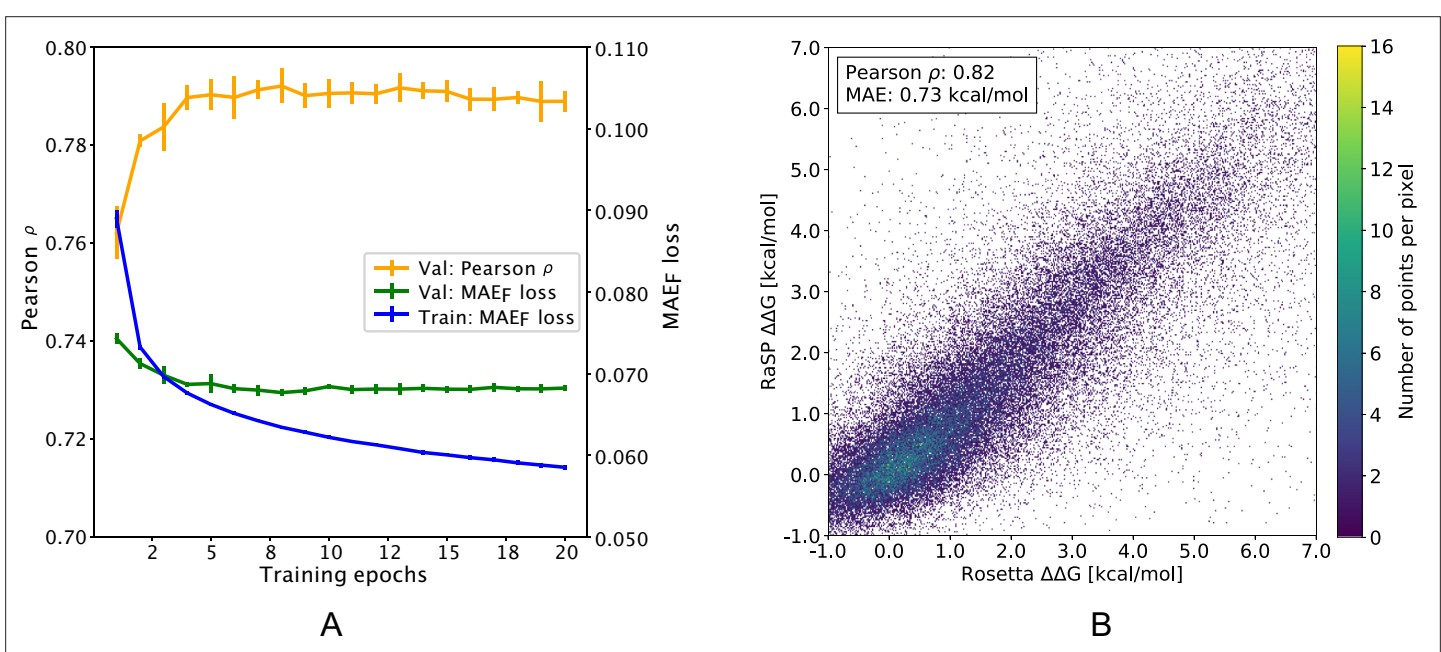

**Figure 2.** Overview of RaSP downstream model training and testing. (**A**) Learning curve for training of the RaSP downstream model, with Pearson correlation coefficients ($\rho$) and mean absolute error ($\mathrm{MAE}_F$) of RaSP predictions. During training we transformed the target $\Delta\Delta G$ data using a switching (Fermi) function, and $\mathrm{MAE}_F$ refers to this transformed data (see Methods for further details). Error bars represent the standard deviation of 10 independently trained models, that were subsequently used in ensemble averaging. Val: validation set; Train: training set. (**B**) After training, we applied the RaSP model to an independent test set to predict $\Delta\Delta G$ values for a full saturation mutagenesis of 10 proteins. Pearson correlation coefficients and mean absolute errors (MAE) were for this figure computed using only variants with Rosetta $\Delta\Delta G$ values in the range [–1;7] kcal/mol.

The online version of this article includes the following figure supplement(s) for figure 2:

**Figure supplement 1.** Learning curve for the self-supervised 3D convolutional neural network.

**Figure supplement 2.** Mean absolute prediction error for RaSP on the validation set, split by amino acid type of the wild-type and variant residue.

**Figure supplement 3.** RaSP versus Rosetta $\Delta\Delta G$ values for a full saturation mutagenesis of 10 test proteins separated into either exposed (**A**) or buried (**B**) residues.

**Table 1.** Overview of RaSP model test set prediction results including benchmark comparison with the Rosetta protocol.

When comparing RaSP to Rosetta (column: "Pearson $|\rho|$ RaSP vs. Ros."), we only compute the Pearson correlation coefficients for variants with a Rosetta $\Delta\Delta G$ value in the range [–1;7] kcal/mol. Experimental data is from *Kumar, 2006*; *Ó Conchúir et al., 2015*; *Nisthal et al., 2019*; *Matreyek et al., 2018*; *Suiter et al., 2020*.

| Data set | Protein name | PDB, chain | Pearson $|\rho|$ RaSP vs. Ros. | Pearson $|\rho|$ RaSP vs. Exp. | Pearson $|\rho|$ Ros. vs. Exp. |
|---|---|---|---|---|---|
| RaSP test set | MEN1 | 3U84, A | 0.85 | - | - |
| | F8 | 2R7E, A | 0.71 | - | - |
| | ELANE | 4WVP, A | 0.81 | - | - |
| | ADSL | 2J91, A | 0.84 | - | - |
| | GCK | 4DCH, A | 0.84 | - | - |
| | RPE65 | 4RSC, A | 0.84 | - | - |
| | TTR | 1F41, A | 0.88 | - | - |
| | ELOB | 4AJY, B | 0.87 | - | - |
| | SOD1 | 2CJS, A | 0.84 | - | - |
| | VANX | 1R44, A | 0.83 | - | - |
| ProTherm test set | Myoglobin | 1BVC, A | 0.91 | 0.71 | 0.76 |
| | Lysozyme | 1LZ1, A | 0.80 | 0.57 | 0.65 |
| | Chymotrypsin inhib. | 2CI2, I | 0.79 | 0.65 | 0.68 |
| | RNAse H | 2RN2, A | 0.78 | 0.79 | 0.71 |
| Protein G | Protein G | 1PGA, A | 0.90 | 0.72 | 0.72 |
| MAVE test set | NUDT15 | 5BON, A | 0.83 | 0.50 | 0.54 |
| | TPMT | 2H11, A | 0.86 | 0.48 | 0.49 |
| | PTEN | 1D5R, A | 0.87 | 0.52 | 0.53 |

environment (see Methods and *Figure 2—figure supplement 1* for further details). In the second step of developing RaSP, we trained a downstream supervised fully-connected neural network to predict stability changes; this model uses the internal representation of the 3D convolutional neural network as well as the corresponding wild type and mutant amino acid labels and frequencies as input. We trained this model on $\Delta\Delta G$ values generated by saturation mutagenesis using Rosetta to limit the effects of biases from experimental assays and amino acid compositions; as described further below we validated the model using a range of different experiments. In the training of the supervised model, we used a custom loss function to focus the model on $\Delta\Delta G$ values in the range from approximately -1 to 7 where Rosetta is known to be most accurate (*Kellogg et al., 2011*) and where many loss-of-function variants relevant for human disease can be identified (*Casadio et al., 2011*; *Cagiada et al., 2021*).

After training (*Figure 2a*), the downstream model achieves a Pearson correlation coefficient of 0.82 and a mean absolute error (MAE) of 0.73 kcal/mol on a test data set comprised of 10 proteins with full saturation mutagenesis which was not seen by the model until after it had been developed and trained (*Figure 2b* and *Table 1*). In practice, the downstream model predictions were made using the median of an ensemble of 10 model predictions with each model trained using different initialization seeds. The accuracy of the model is relatively uniform for the different types of amino acid substitutions, with bigger errors in particular when substituting glycine residues, or when changing residues to proline (*Figure 2—figure supplement 2*). In terms of location of the residue in the protein structure, the model accuracy is slightly better at exposed relative to buried residues (*Figure 2—figure*

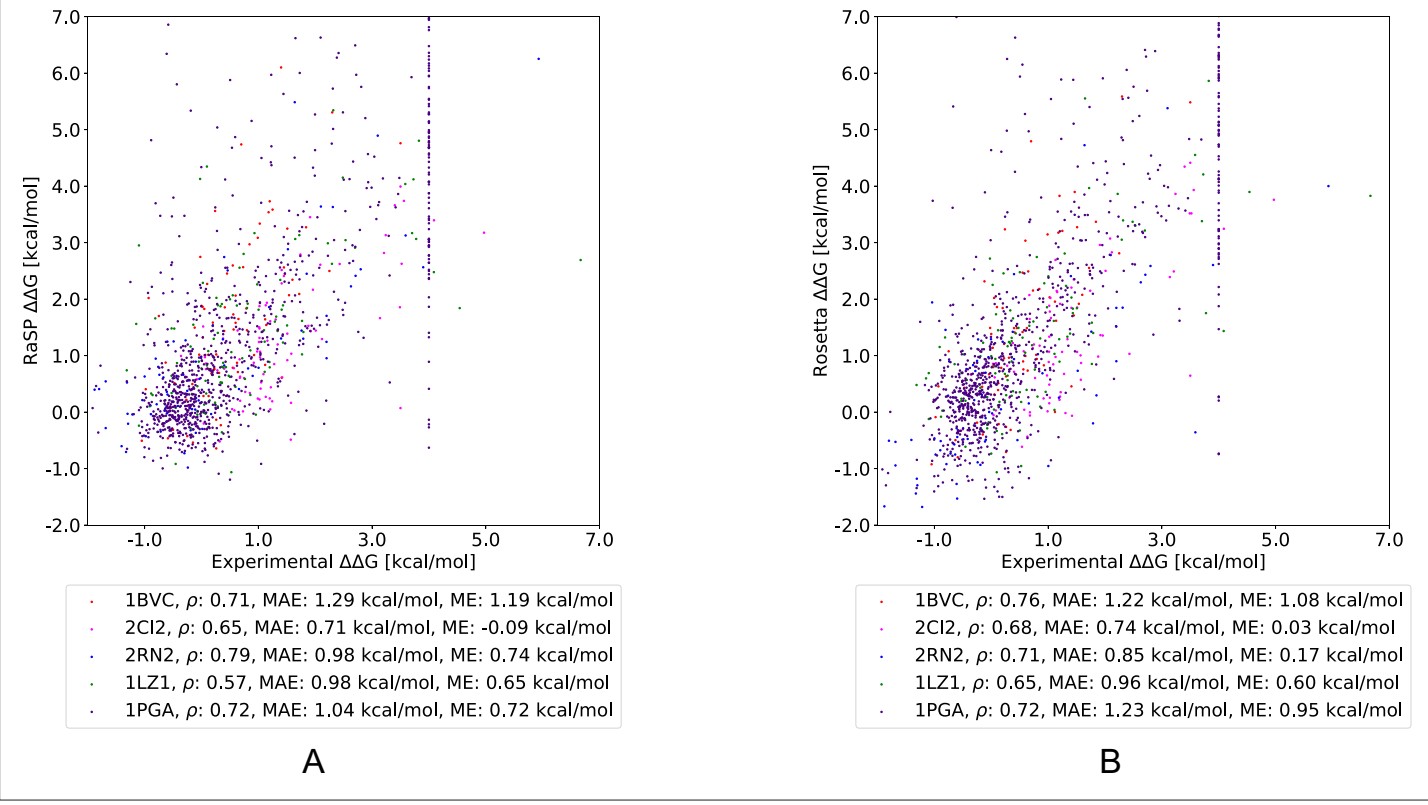

**Figure 3.** Comparing RaSP and Rosetta predictions to experimental stability measurements. Predictions of changes in stability obtained using (**A**) RaSP and (**B**) Rosetta are compared to experimental data on five test proteins; myoglobin (1BVC), lysozyme (1LZ1), chymotrypsin inhibitor (2CI2), RNAse H (2RN2) and Protein G (1PGA) (***Kumar, 2006***; ***Ó Conchúir et al., 2015***; ***Nisthal et al., 2019***). Metrics used are Pearson correlation coefficient ($\rho$), mean absolute error (MAE) and mean error (ME). In the experimental study of Protein G, 105 variants were assigned a $\Delta\Delta G$ value of at least 4 kcal/mol due to low stability, presence of a folding intermediate, or lack expression (***Nisthal et al., 2019***).

The online version of this article includes the following figure supplement(s) for figure 3:

**Figure supplement 1.** Comparing RaSP and Rosetta predictions to experimental stability measurements.

**Figure supplement 2.** RaSP performance on three recently published data sets (***Pancotti et al., 2022***): (**A**) The S669 data set, (**B**) The Ssym+ direct data set, (**C**) The Ssym+ reverse data set.

**Figure supplement 3.** RaSP performance on the recently published mega-scale experiments (***Tsuboyama et al., 2022***).

**Figure supplement 4.** Benchmarking RaSP and Rosetta using VAMP-seq data.

supplement 3), although we note also that the distribution of stability changes are different in the two different locations.

## Validating RaSP using experimental data

After having shown that the RaSP model can successfully reproduce Rosetta $\Delta\Delta G$ values, we proceeded to test it on experimental measurements for five proteins. Specifically, we selected four proteins with stability measurements from ProTherm (***Kumar, 2006***; ***Ó Conchúir et al., 2015***) and the B1 domain of protein G for which stability measurements are available for almost all possible single amino acid substitutions (***Nisthal et al., 2019***). We compared RaSP predictions for the five test proteins to Rosetta as a baseline (***Figure 3***, ***Figure 3—figure supplement 1***). Pearson correlation coefficients of RaSP model predictions versus experimental stability measurements ranged from 0.79 (RNAse H, Rosetta baseline: 0.71)–0.57 (lysozyme, Rosetta baseline: 0.65). Although these correlations are not very high on an absolute scale, our results indicate that the RaSP model is approximately as accurate as the Rosetta protocol for predicting experimental $\Delta\Delta G$ values (***Table 1***). We also note that this accuracy is close to what is computationally achievable given the available data; it has been shown that there exists a natural upper bound on the achievable accuracy when predicting experimental

**Table 2.** Benchmark performance of RaSP versus other structure-based methods on the S669 direct experimental data set (*Pancotti et al., 2022*).

Results for methods other than RaSP have been copied from *Pancotti et al., 2022*. We speculate that the higher RMSE and MAE values for Rosetta relative to RaSP are due to missing scaling of Rosetta output onto a scale similar to kcal/mol.

| Method | Pearson$\rho$ | S669, direct RMSE [kcal/mol] | MAE [kcal/mol] |
|---|---|---|---|
| *Structure-based* | | | |
| ACDC-NN | 0.46 | 1.49 | 1.05 |
| DDGun3D | 0.43 | 1.60 | 1.11 |
| PremPS | 0.41 | 1.50 | 1.08 |
| **RaSP** | 0.39 | 1.63 | 1.14 |
| ThermoNet | 0.39 | 1.62 | 1.17 |
| Rosetta | 0.39 | 2.70 | 2.08 |
| Dynamut | 0.41 | 1.60 | 1.19 |
| INPS3D | 0.43 | 1.50 | 1.07 |
| SDM | 0.41 | 1.67 | 1.26 |
| PoPMuSiC | 0.41 | 1.51 | 1.09 |
| MAESTRO | 0.50 | 1.44 | 1.06 |
| FoldX | 0.22 | 2.30 | 1.56 |
| DUET | 0.41 | 1.52 | 1.10 |
| I-Mutant3.0 | 0.36 | 1.52 | 1.12 |
| mCSM | 0.36 | 1.54 | 1.13 |
| Dynamut2 | 0.34 | 1.58 | 1.15 |

$\Delta\Delta G$ values due to variations between experiments (*Montanucci et al., 2019*). Furthermore, we find the bias of the predictions (assessed via the mean error) is roughly the same between RaSP and Rosetta.

In order to further validated RaSP, we compared its performance versus other methods on the recently published S669 experimental direct data set (*Pancotti et al., 2022*; *Figure 3—figure supplement 2*). The data set includes 669 variants and 94 experimental structures. We observe that RaSP performs as well as Rosetta in terms of Pearson correlation on this particular data set (*Figure 3—figure supplement 2*) and that RaSP performs relatively close to several of the best performing methods (*Table 2*). We also test the performance of RaSP on a high-accuracy version of the recently published experimental mega-scale data set from Rocklin and collaborators (*Tsuboyama et al., 2022*). We observe that RaSP achieves a zero-shot Pearson correlation coefficient of 0.62 and an MAE of 0.94 (*Figure 3—figure supplement 3*). Taken together, these results suggest the RaSP performs on-par with other computational methods although differences in training and parameterization makes direct comparison difficult. For example, a given method might have been parameterized either directly or independently from experimental data. In the case of RaSP, parameterization is made indirectly from experimental data since the Rosetta $\Delta\Delta G$ itself has been parameterized using a range of different types of experimental data (*Park et al., 2016*).

A common challenge for many machine learning-based methods is the ability to satisfy the anti-symmetry condition, which states that the $\Delta\Delta G$ of a single point mutation must equal its reverse mutation with opposite sign (*Pancotti et al., 2022*). In order to assess this property, we tested RaSP on the Ssym+ data set, which includes both a direct and a reverse data set (*Pancotti et al., 2022*). The direct data set includes 352 variants and 19 experimental structures while the reverse data set includes 352 experimental structures for each of the corresponding reverse variants. RaSP achieves a Pearson correlation of 0.58 and a MAE of 1.01 kcal/mol on the direct data set, while it only achieves a Pearson correlation of 0.18 and a MAE of 1.82 kcal/mol on the reverse data set (*Figure 3—figure supplement 2*). The failure of RaSP to accurately capture anti-symmetric effects is expected following the one-sided training method and provides an interesting avenue for further method development.

Multiplexed assays of variant effects (MAVEs, also known as deep mutational scanning assays) leverage recent developments in high-throughput DNA synthesis and sequencing to probe large (e.g. thousands per protein) libraries of protein variants (*Kinney and McCandlish, 2019*). A particular type of MAVE termed 'Variant Abundance by Massively Parallel sequencing' (VAMP-seq) probes variant effects on cellular protein abundance (*Matreyek et al., 2018*), and correlates with both in vitro measurements (*Matreyek et al., 2018*; *Suiter et al., 2020*) and computational predictions (*Cagiada et al., 2021*) of protein stability. We therefore compared RaSP calculations with VAMP-seq data for

three proteins and find that it correlates about as well as Rosetta calculations (*Table 1* and *Figure 3—figure supplement 4*).

## Prediction of protein stability-change using computationally modelled structures

For maximal utility for proteome-wide stability predictions, our model should be generally robust to the quality of input protein structures (*Caldararu et al., 2021*). To test this, we used template-based (homology) modelling to generate structures of the four proteins selected from ProTherm that we analysed above. To minimize issues of leakage between training and testing data, we have recently used MODELLER (*Martí-Renom et al., 2000*; *Webb and Sali, 2016*) to construct models of the four proteins, using templates with decreasing sequence identities to the original structures (*Valanciute et al., 2023*). We used these structures as input to RaSP in order to calculate $\Delta\Delta G$ values to compare with experiments; we also performed similar tests using $\Delta\Delta G$ values calculated using Rosetta (*Figure 4*). Overall we find that both RaSP and Rosetta are relatively insensitive to the input models, with some decrease in accuracy for both methods when models are created using more distant homologues.

The development of AlphaFold 2 (*Jumper et al., 2021*) and other protein structure prediction algorithms has enabled large-scale generation of accurate structural models (*Varadi et al., 2022*). Recently, it has been shown that high-confidence models generated by AlphaFold 2 are sufficiently accurate to be used as input for protein stability predictions using for example FoldX and Rosetta (*Akdel et al., 2022*). We therefore tested whether AlphaFold 2 structures could similarly be used as input to RaSP. More specifically, we asked whether $\Delta\Delta G$ could be predicted as accurately from AlphaFold 2 structures as from crystal structures. We selected three proteins that each had two high-resolution crystal structures and used these as well as an AlphaFold 2 model to predict $\Delta\Delta G$. Overall we find a high correlation between the RaSP predictions from the pairs of crystal structures ($\overline{\rho} = 0.97$), only slightly greater than the correlation between values predicted from a crystal structure and an AlphaFold 2 structure ($\overline{\rho} = 0.94$) (*Table 3*). This correlation is highest in regions of the proteins where AlphaFold 2 is most confident about the structure prediction (i.e. residues with high pLDDT scores). Interpreting this observation is, however, complicated by the fact that low pLDDT scores are often found in loop regions that might have naturally lower $\Delta\Delta G$ scores, thus increasing the effects of noise and minor outliers on correlation scores. Overall, these results, together with those using the homology models above, indicate that RaSP can also deliver accurate predictions using computationally generated protein structures; we note that similar results have been obtained using other stability prediction methods (*Akdel et al., 2022*; *Valanciute et al., 2023*; *Lihan et al., 2023*; *Keskin Karakoyun et al., 2023*).

## Large-scale calculations of stability-changes and analysis of disease-causing missense variants

Protein stability has been shown to be an important driver of human disease (*Casadio et al., 2011*; *Martelli et al., 2016*; *Nielsen et al., 2020*) and knowledge about changes in protein stability can provide important mechanistic insights into the molecular origins of a disease (*Abildgaard et al., 2023*; *Cagiada et al., 2021*). In particular, predictions of changes in protein stability can help separate variants into mechanistic classes depending on how they affect protein function (*Cagiada et al., 2022*). In this light, developing accurate predictors of $\Delta\Delta G$ values is an important step in understanding the mechanisms behind human diseases.

Large-scale predictions often require models that are specifically suited for the task. As such, we designed RaSP to be fast, yet retain the accuracy of the Rosetta protocol. We benchmarked the speed of RaSP against both Rosetta and FoldX (well-established energy-function-based methods) (*Kellogg et al., 2011*; *Schymkowitz et al., 2005*) as well as two recent machine learning-based methods; ACDC-NN and ThermoNet (*Benevenuta et al., 2021*; *Li et al., 2020*; *Table 4*). We find that RaSP enables saturation mutagenesis $\Delta\Delta G$ calculations at a speed of ~0.4 s per position—independent of total protein length—after pre-processing of the protein structure. At this speed, saturation mutagenesis scans are possible for most proteins in a few minutes—orders of magnitude faster than Rosetta (*Table 4*; note differences in hardware used for the two sets of calculations). Furthermore, we find that RaSP is also faster than the other machine learning-based methods that have not been optimized for

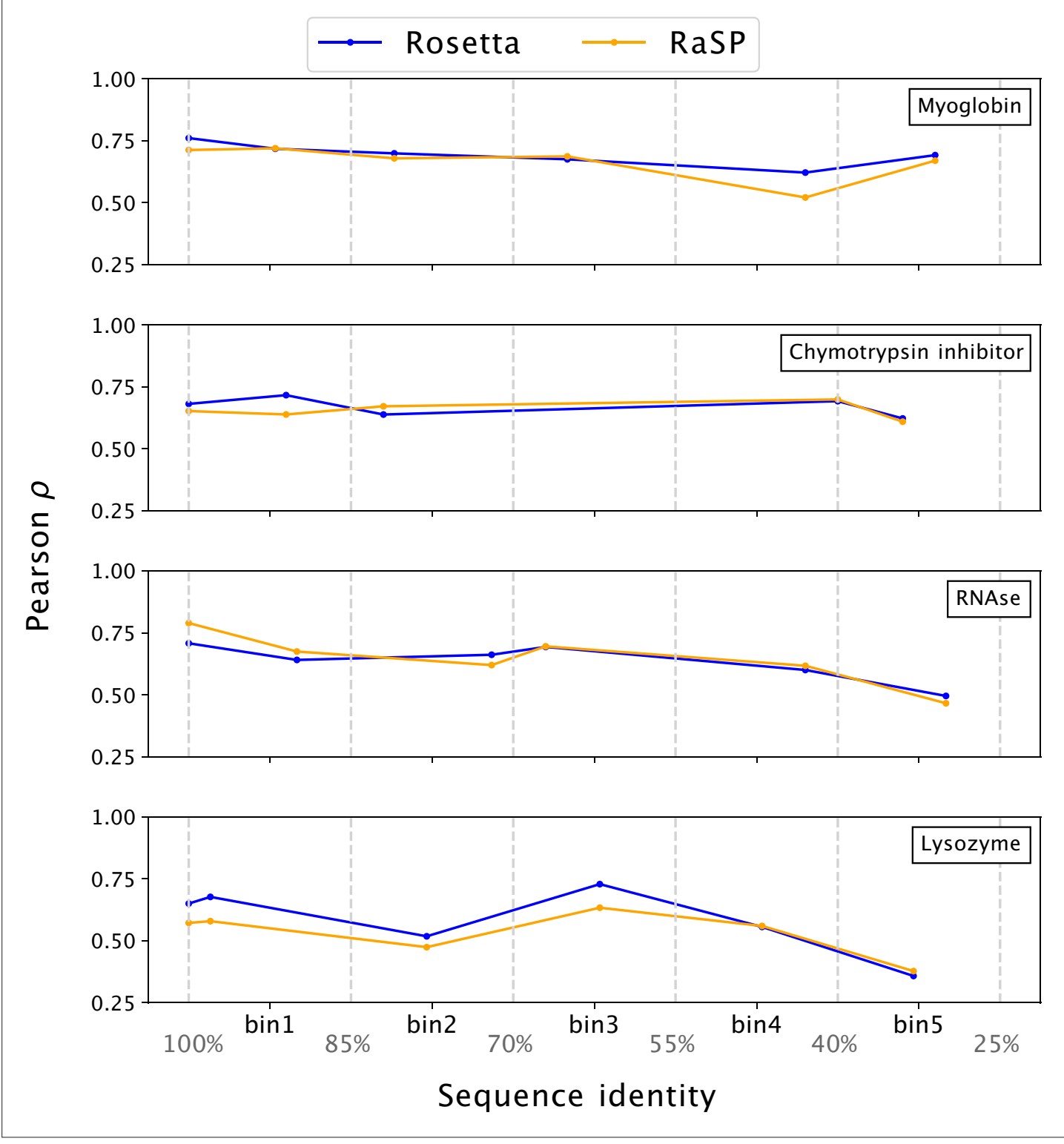

**Figure 4.** Stability predictions from structures created by template-based modelling. Pearson correlation coefficients ($\rho$) between experimental stability measurements and predictions using protein homology models with decreasing sequence identity to the target sequence. Pearson correlation coefficients were computed in the range of [–1;7] kcal/mol.

**Table 3.** Comparing RaSP predictions from crystal and AlphaFold 2 (AF2) structures. Pearson correlation coefficients ($\rho$) between RaSP $\Delta\Delta G$ predictions using either two different crystal structures or a crystal structure and an AlphaFold 2 structure for six test proteins: PRMT5 (X1: 6V0P_A, X2: 4GQB_A), PKM (X1: 6B6U_A, X2: 6NU5_A), FTH1 (X1: 4Y08_A, X2: 4OYN_A), FTL (X1: 5LG8_A, X2: 2FFX_J), PSMA2 (X1: 5LE5_A, X2: 5LE5_O) and GNB1 (X1: 6CRK_B, X2: 5UKL_B). We also divided the analysis into residues with high (pLDDT ≥ 0.9) and medium-low (pLDDT <0.9) pLDDT scores from AlphaFold 2.

| Protein | All [$\rho$] | | | High AF2 pLDDT [$\rho$] | | | Medium-Low AF2 pLDDT [$\rho$] | | |
|---|---|---|---|---|---|---|---|---|---|
| | X1-X2 | X1-AF2 | X2-AF2 | X1-X2 | X1-AF2 | X2-AF2 | X1-X2 | X1-AF2 | X2-AF2 |
| PRMT5 | 0.93 | 0.89 | 0.95 | - | 0.90 | 0.95 | - | 0.66 | 0.89 |
| PKM | 0.99 | 0.95 | 0.95 | - | 0.95 | 0.95 | - | 0.88 | 0.89 |
| FTH1 | 0.99 | 0.97 | 0.97 | - | 0.97 | 0.97 | - | 0.92 | 0.95 |
| FTL | 0.97 | 0.96 | 0.97 | - | 0.96 | 0.97 | - | 0.96 | 0.94 |
| PSMA2 | 0.99 | 0.95 | 0.95 | - | 0.96 | 0.96 | - | 0.78 | 0.80 |
| GNB1 | 0.96 | 0.94 | 0.94 | - | 0.94 | 0.94 | - | 0.93 | 0.89 |

**Table 4.** Run-time comparison of RaSP and four other methods for three test proteins ELOB (PDB: 4AJY_B, 107 residues), GCK (PBD: 4DCH_A, 434 residues) and F8 (PDB: 2R7E_A, 693 residues). The RaSP model is in total 480–1,036 times faster than Rosetta. RaSP, ACDC-NN and ThermoNet computations were performed using a single NVIDIA V100 16 GB GPU machine, while Rosetta and FoldX computations were parallelized and run on a server using 64 2.6 GHz AMD Opteron 6380 CPU cores. The number of $\Delta\Delta G$ computations per mutation was set to 3 for both Rosetta and FoldX. For ThermoNet, we expect that the pre-processing speed can be made comparable to Rosetta via parallelization.

| Method | Protein | Wall-clock time [s] | | |
|---|---|---|---|---|
| | | Pre-processing | $\Delta\Delta G$ | $\Delta\Delta G$/ residue |
| RaSP | ELOB | 7 | 41 | 0.4 |
| | GCK | 11 | 173 | 0.4 |
| | F8 | 20 | 270 | 0.4 |
| Rosetta | ELOB | 677 | 44,324 | 414.2 |
| | GCK | 7,996 | 118,361 | 272.7 |
| | F8 | 17,211 | 133,178 | 192.2 |
| FoldX | ELOB | 78 | 42,237 | 394.7 |
| | GCK | 728 | 309,762 | 713.7 |
| | F8 | 1,306 | 559,050 | 806.7 |
| ACDC-NN | ELOB | 81 | 158 | 1.5 |
| | GCK | 169 | 619 | 1.4 |
| | F8 | 325 | 1,080 | 1.6 |
| ThermoNet | ELOB | 80,442 | 884 | 8.3 |
| | GCK | 4,586,522 | 4,227 | 9.7 |
| | F8 | 11,627,433 | 8,732 | 12.6 |

large-scale applications. The ACDC-NN model is the closest performing model to RaSP in terms of speed, and ACDC-NN has also shown to have good predictive performance on experimental test data (*Table 2*). ACDC-NN is, however, somewhat slower than RaSP both in time for pre-processing and for computing $\Delta\Delta G$; it also uses information from multiple sequence alignments, which might not always be available.

To demonstrate how the speed of RaSP enables large-scale applications, we used it to perform saturation scans for 1381 experimentally determined structures of human proteins or domains (see Methods for further details on selection criteria), yielding a data set of ~ 8.8 million $\Delta\Delta G$ values. The $\Delta\Delta G$ values predicted by RaSP follow the expected distribution (*Tokuriki et al., 2007*; *Nisthal et al., 2019*) with a small number of stabilizing variants, a large number of slightly destabilizing variants and a long tail of very destabilizing variants (*Figure 5—figure supplement 1*).

As an example of the kinds of analyses that these large-scale calculations enable, we examined variants that have been observed in the human population. Specifically, we extracted validated disease-causing and benign variants from ClinVar (*Landrum et al., 2018*) and variants observed more broadly in the human population from the Genome Aggregation Database (gnomAD) (*Karczewski et al., 2020*), and annotated them with their predicted $\Delta\Delta G$ values (*Figure 5*). We find that benign variants generally only have a smaller effect on stability (median $\Delta\Delta G$ 0.54 kcal/mol, and 95% within the range –0.9–2.7 kcal/mol; *Figure 5A*) whereas many disease causing missense variants are predicted to be destabilizing (median $\Delta\Delta G$ 1.4 kcal/mol, and 95% within the range –1.4–6.7 kcal/mol; *Figure 5A*). Thus, a substantial fraction of the pathogenic variants are predicted to be destabilized to an extent that is likely to cause lowered abundance (*Cagiada et al., 2021*; *Høie et al., 2022*). We observe that this difference in median $\Delta\Delta G$ values between the benign and the pathogenic group is statistically significant using bootstrap sampling. We resample each of the pathogenic and benign distributions with replacement $10^4$ times and compute the difference in medians between the two groups along with a 95% confidence interval (CI). Using this method, the difference in medians is estimated at 0.82 kcal/mol (CI: 0.73 kcal/mol – 0.93 kcal/mol).

Similarly, variants that are common in the human population (i.e. with an allele frequency in gnomAD > $10^{-2}$) generally only have a small effect on the stability (median $\Delta\Delta G$ 0.41 kcal/mol), whereas rarer variants have a broader distribution including more destabilizing variants (*Figure 5B*). Thus, in line with previous observations (*Casadio et al., 2011*; *Martelli et al., 2016*; *Stein et al., 2019*) we find that loss of protein stability is likely to be the underlying cause for loss of function for a large fraction of pathogenic variants. Using the bootstrapping method described above, we observe that the difference in median $\Delta\Delta G$ values between the common (gnomAD > $10^{-2}$) and the rare (gnomAD < $10^{-4}$) variants is estimated at 0.35 kcal/mol (CI: 0.27 kcal/mol – 0.43 kcal/mol).

Following the prediction of ~8.8 million RaSP $\Delta\Delta G$ values, we decided to expand our analysis to the entire human proteome corresponding to ~ 300 million predicted RaSP $\Delta\Delta G$ values across 23,391 protein structures obtained from the AlphaFold2 database (*Varadi et al., 2022*; *Figure 5—figure supplement 3*); because of overlapping fragments in AlphaFold predictions for large proteins, this corresponds to ca. 230 million unique variants. We note that RaSP has only been trained and benchmarked on soluble, globular and folded proteins and may therefore not expected to perform equally well on proteins outside this category including e.g. membrane proteins and intrinsically disordered proteins. Furthermore, AlphaFold2 may not predict reliable structural models in regions that are highly flexible and disordered (*Jumper et al., 2021*; *Varadi et al., 2022*). For convenience, we however provide the data for all human proteins. Despite these caveats, we observe that the RaSP predictions follow the same expected distributional shape as described above. We believe that these predictions could serve as a fruitful starting point for large-scale studies of protein stability and have therefore made these RaSP predictions freely available online (see 'Data and code availability').

## Discussion

Here, we have presented a novel deep-learning-based model for making rapid predictions of protein stability changes using a combination of supervised and self-supervised methods. Our model uses the latent-space representation of a 3D convolutional neural network for protein structures as input to a supervised model to predict $\Delta\Delta G$. Our results show that the RaSP model performs on-par with the Rosetta protocol that we used as target for our optimization, but at a speed that is several orders of magnitude faster. We analysed RaSP's sensitivity towards using predicted protein structures, and

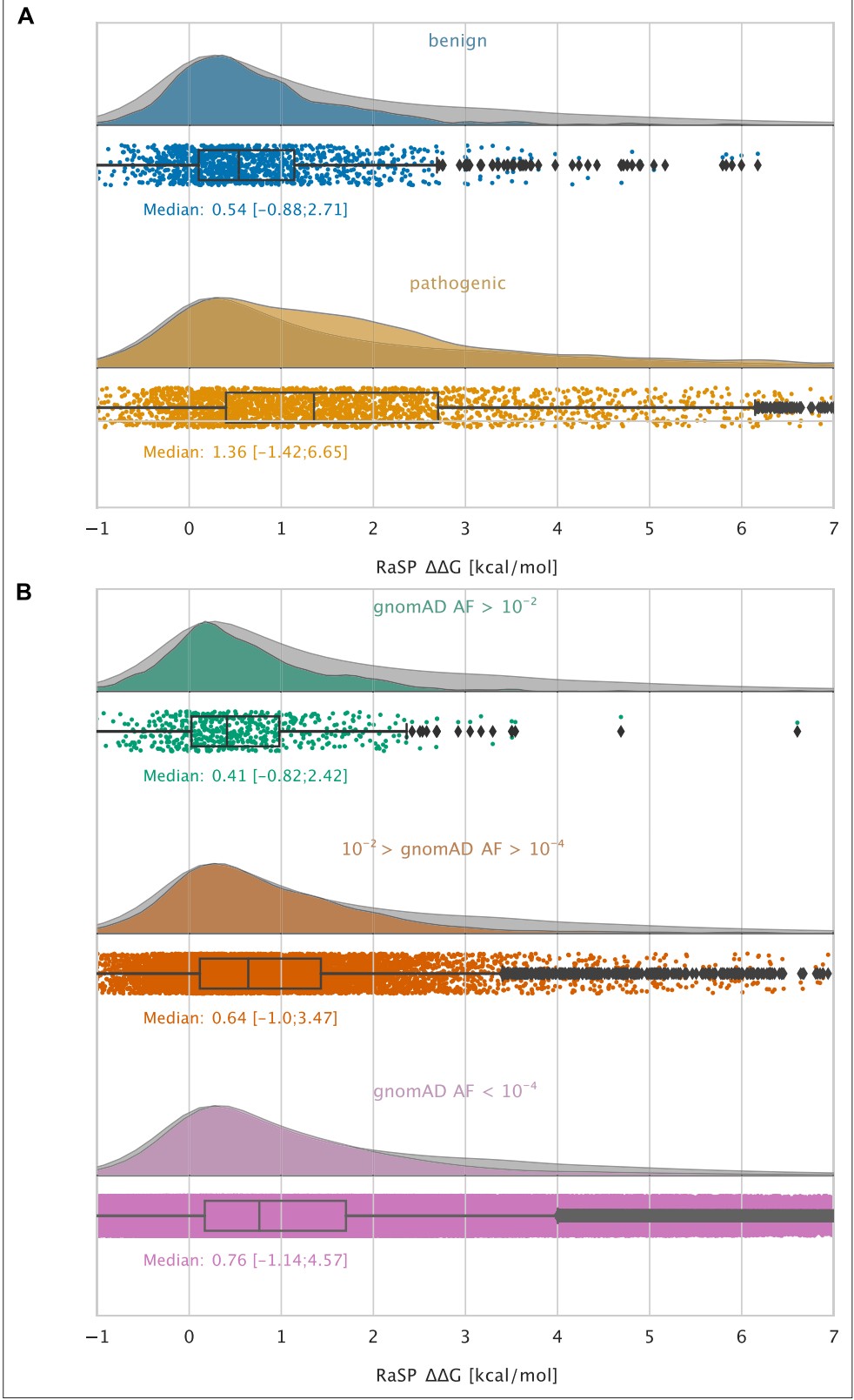

**Figure 5.** Large-scale analysis of disease-causing variants and variants observed in the population. The grey distribution shown in the background of all plots represents the distribution of $\Delta\Delta G$ values calculated using RaSP for all single amino acid changes in the 1,366 proteins that we analysed (15 of the 1381 proteins that we calculated $\Delta\Delta G$ for did not have variants in ClinVar or gnomAD and were therefore not included in this analysis). Each plot

*Figure 5 continued on next page*

*Figure 5 continued*

is also labelled with the median $\Delta\Delta G$ of the subset analysed as well as a range of $\Delta\Delta G$ values that cover 95% of the data in that subset (box plot shows median, quartiles and outliers). The plots only show values between –1 and 7 kcal/mol (for the full range see *Figure 5—figure supplement 2*). (**A**) Distribution of RaSP $\Delta\Delta G$ values for benign (blue) and pathogenic (tan) variants extracted from the ClinVar database (*Landrum et al., 2018*). We observe that the median RaSP $\Delta\Delta G$ value is significantly higher for pathogenic variants compared to benign variants using bootstrapping. (**B**) Distribution of RaSP $\Delta\Delta G$ values for variants with different allele frequencies (AF) extracted from the gnomAD database *Karczewski et al., 2020* in the ranges (**i**) AF>$10^{-2}$ (green), (ii) $10^{-2}$ > AF>$10^{-4}$ (orange), and (iii) AF<$10^{-4}$ (purple). We observe a gradual shift in the median RaSP $\Delta\Delta G$ going from common variants (AF>$10^{-2}$) towards rarer ones (AF<$10^{-4}$).

The online version of this article includes the following figure supplement(s) for figure 5:

**Figure supplement 1.** Histogram of $\Delta\Delta G$ values from saturation mutagenesis using RaSP on 1,366 PDB structures corresponding to ~8.8 million predicted $\Delta\Delta G$ values.

**Figure supplement 2.** Large-scale analysis of disease-causing variants and variants observed in the population using the RaSP model.

**Figure supplement 3.** Histogram of $\Delta\Delta G$ values from saturation mutagenesis using RaSP on predicted structures of the entire human proteome corresponding to ~300 million predicted $\Delta\Delta G$ values predicted from 23,391 protein structures.

---

applied it to generate a unique and large set of stability-change predictions for 8.8 million variants of 1,381 human proteins. As an example of the kinds of applications that RaSP may enable, we analysed the role of protein stability for genetic diseases, and hope that this data and the RaSP model will be a broadly useful tool to study the effects of amino acid changes on protein stability.

## Methods
### A convolutional neural network for protein structures

We use a 3D convolutional neural network as a model to represent protein structures based on the architectures presented in *Boomsma and Frellsen, 2017*. During training, the model is tasked with predicting the amino acid type given its local 3D atomic environment. Specifically, for each residue in the polypeptide chain, the model aims to predict the amino acid type based on the local atomic environment lying inside a sphere centered on the $C_\alpha$ atom of the target residue to be predicted with a radius of 9 Å. Before prediction, all atoms in the target residue are removed from the environment. Each local spherical environment is parameterized using a Cartesian grid and so divided into multiple cubic voxels each with a volume of $(1\mathring{A})^3$. Each cubic voxel has six input channels corresponding to the one-hot encoding of C, N, O, H, S, and P atoms. The model consists of a series of three 3D convolutional layers that is applied to the local spherical atomic environment. All convolutional filters have a size of $(3\mathring{A})^3$ and the number of filters at each convolutional layer is 16, 32, and 64, respectively. The three 3D convolutional layers are connected using leaky ReLU activation (*He et al., 2015*), max pooling (*Riesenhuber and Poggio, 1999*) and batch normalization functions (*Ioffe and Szegedy, 2015*). We apply Gaussian blurring to the input local environment before the convolutional layers. After the convolutional layers, the output is flattened and the dimensionality is reduced by two fully connected layers with 100 and 20 nodes, respectively. Model predictions are optimized by minimizing the cross-entropy between the predicted 20-length vector and the one-hot encoded amino acid label.

We trained the above model on a data set of 2336 high resolution protein structures obtained using the PISCES server with maximal sequence identity set to 30% (*Wang and Dunbrack, 2003*). All structures were pre-processed using the Reduce program (*Word et al., 1999*) to add missing hydrogen atoms and OpenMM PDBFixer (*Eastman et al., 2013*) to correct for common PDB errors; during pre-processing, three proteins gave errors and were thus discarded. Out of the remaining 2333 proteins, 2099 proteins were used for training while 234 were withheld for validation. Optimization was performed using Adam (*Kingma and Ba, 2014*) with a learning rate of $3 \cdot 10^{-4}$ and a batch size of 100 residues. After training, the representation model achieves a wild-type amino acid classification accuracy of 63% on a validation set (*Figure 2—figure supplement 1*).

## Downstream model architecture and training

We used the convolutional neural network described above to represent protein structures when predicting protein stability changes. Specifically, for each variant, the downstream model takes as input (i) a flattened 100-dimensional representation of the atomic environment extracted for the wild-type structure using the self-supervised model, (ii) one-hot encoded labels of the wild type and variant amino acid, and (iii) the amino acid frequencies of the wild type and variant amino acid in the PISCES set of proteins. These inputs are concatenated and passed through three fully connected layers with 128, 64, and 16 nodes, respectively, resulting in a scalar value. During training, this value is passed through a sigmoid function corresponding to the transformation of the target data described below. Between each layer, we apply batch normalization (*Ioffe and Szegedy, 2015*) and leaky ReLU activation (*He et al., 2015*). We train the model by minimizing the mean absolute error between the predicted values of the stability-change and Rosetta $\Delta\Delta G$ values, after being transformed using a switching function (see below).

We trained the downstream model using $\Delta\Delta G$ values from saturation mutagenesis of 35 training proteins and 10 validation proteins, and tested using 10 different proteins. Before training, we transformed all Rosetta $\Delta\Delta G$ values using a switching (Fermi) function to focus model training on the $\Delta\Delta G$ range from -1 to 7, as this is where Rosetta is expected to be most accurate (*Kellogg et al., 2011*), where most experimental values lie (*Kumar, 2006*; *Ó Conchúir et al., 2015*; *Nisthal et al., 2019*), and covering an important range for detecting missense variants that cause disease via loss of protein stability and abundance (*Stein et al., 2019*; *Cagiada et al., 2021*; *Høie et al., 2022*). The Fermi function was defined as:

$$F(\Delta\Delta G) = \frac{1}{1 + e^{-\beta(\Delta\Delta G - \alpha)}} \qquad (1)$$

with $\beta = 0.4$ and $\alpha = 3.0$. Optimization was performed using Adam (*Kingma and Ba, 2014*) with a learning rate of $5 \cdot 10^{-4}$ and a batch size of 40 variants. Stability changes for (disulfide-bonded) cystine residues were removed from the training, validation and test set as they are not predicted using our Rosetta protocol. In practice, this means that RaSP will tend to underestimate the magnitude of $\Delta\Delta G$ values for mutations at cysteine residues involved in disulfide-bonds and that these predicted $\Delta\Delta G$ values should therefore not be taken to be accurate.

## Rosetta protocol

We used Rosetta (GitHub SHA1 99d33ec59ce9fcecc5e4f3800c778a54afdf8504) and the the Cartesian $\Delta\Delta G$ protocol (*Park et al., 2016*) to predict $\Delta\Delta G$ values. Values from Rosetta values were divided by 2.9 to convert to a scale corresponding to kcal/mol (*Park et al., 2016*).

## Calculation of SASA

We analyse the differences in correlation between RaSP and Rosetta $\Delta\Delta G$ predictions at exposed and buried residues respectively (*Figure 2—figure supplement 3*). The solvent accessible surface area (SASA) was calculated using BioPython and default Sander and Rost values (*Cock et al., 2009*; *Rost and Sander, 1994*). The cut-off between exposed and buried residues was set at 0.2.

## Filtering of mega-scale data set

We tested the performance of RaSP on a recently published mega-scale folding stability experiment (*Tsuboyama et al., 2022*). We included variants with single amino acid substitutions with well-defined experimental $\Delta\Delta G$ values. As RaSP has only been trained on $\Delta\Delta G$ values from natural protein structures, we also decided to include only those in our test. Furthermore, we excluded any synonymous substitutions. The filtered data set contains a total of 164,524 variants across 164 protein domain structures.

## Speed test benchmarking

We compare the speed of RaSP to other predictors of $\Delta\Delta G$ namely Rosetta, FoldX, ACDC-NN and ThermoNet (*Kellogg et al., 2011*; *Schymkowitz et al., 2005*; *Benevenuta et al., 2021*; *Li et al., 2020*). The Rosetta protocol was implemented as described above. The FoldX protocol was implemented using FoldX version 5.0 with a custom script to enable full saturation mutagenesis calculations.

ACDC-NN was implemented using the GitHub source code (https://github.com/compbiomed-unito/acdc-nn, *Computational Biology and Medicine Group, 2023* accessed January 2023). ThermoNet was implemnted using the GitHub source code (https://github.com/gersteinlab/ThermoNet, *Gerstein Lab, 2022* accessed January 2023).

## Selection of proteins and structures for large-scale analysis

Our selection of experimental structures was based on 5,557 human proteins annotated with cytosolic location. To make this list, we extracted UniProt identifiers from the human genome assembly GRCh38. p13 filtered for the Gene Ontology (*Ashburner et al., 2000*) term 'cytosol' (GO:0005829). We aligned the sequences of all chains of all PDB entries in the SIFTS map (*Dana et al., 2019*) (June 2021) to the UniProt sequences for the cytosolic proteins and calculated how much the protein structure in each PDB file covered the UniProt entries (explicitly considering mismatches, indels, residues that are not resolved in the structure, and modified residues). Based on these features, we calculated a score for each protein structure and the highest scoring structure was selected to represent a PDB entry. When calculating the score, we gave the highest priority to the coverage of the UniProt sequence and the experimental method. Specifically, given that RaSP was initially developed and benchmarked using crystal structures, we prioritized methods in the following order: (i) X-ray, (ii) cryo-electron microscopy, (iii) NMR. For our final selection, we chose 1,381 structures from the above selection, where the structure was of high quality in a canonical PDB format and where at least 50% of the UniProt sequence was covered by the structure.

## Benchmarking on the S669 dataset

We used RaSP to predict stability changes for 669 variants using 94 experimental structures from a recently described dataset (*Pancotti et al., 2022*). We note that one of the proteins in the S669 set (1XXN.pdb) is highly similar to one of the proteins we used to train our downstream model on (1C5H. pdb). We also used the Ssym+ data set (*Pancotti et al., 2022*), which includes 352 variants and 19 experimental structures for the 'forward' substitutions, and 352 experimental structures for each of the corresponding reverse variants.

## Selection of ClinVar and gnomAD variants

The extraction of variants from ClinVar and gnomAD is based on *Tiemann et al., 2023*. Briefly, we extracted data from gnomAD using an in-house database (scripts available at: https://github.com/KULL-Centre/PRISM/tree/main/software/make_prism_files, *Linderstrøm-Lang Centre for Protein Science, University of Copenhagen, 2021* release-tag v0.1.1). This was constructed using exome data from gnomAD v2 and whole-genome data from gnomAD v3, selecting exome GRCh38 liftover for v2 and whole-genome files for v3, and annotated with Variant Effect Predictor with the GRCh38 release 100 human transcripts set from Ensembl. As previously described (*Tiemann et al., 2023*), we computed gnomAD allele frequencies for all protein variants from the extracted exome- and genome data combined (if present, otherwise solely for the exome- or genome allele frequencies). The reported allele frequencies are calculated as the sum of exome- and genome allele counts divided by the sum of total exome- and genome allele counts, and all DNA variants leading to the same protein-level variant were combined (*Tiemann et al., 2023*). Similarly, ClinVar data were extracted from an in-house database constructed using the NCBI data source: here (May 2021). For our analysis, only single nucleotide variants with a mapping to GRCh38 and with a rating of at least one star were used. We did not find any variants in gnomAD or ClinVar for 15 of the 1381 proteins that we calculated $\Delta\Delta G$ for; the data shown in *Figure 5* thus refers only to 1366 proteins.

## Selection of data for human proteome analysis

The structures used for the human proteome analysis were obtained from the AlphaFold2 database UP000005640_9606_HUMAN_v2 (*Varadi et al., 2022*; *Figure 5—figure supplement 3*). The structures in the database were pre-processed following the RaSP protocol using Reduce and PDBFixer, but were otherwise unmodified so that each protein is defined by a single UniProt sequence and a single predicted structure. For longer proteins the AlphaFold2-predicted structures have been split into fragments; for those proteins we also split our RaSP predictions into corresponding fragments.

## Data and code availability

Scripts and data to repeat our analyses are available via: https://github.com/KULL-Centre/_2022_ML-ddG-Blaabjerg/, where we also provide a link to run RaSP via resources at Google Colaboratory.

## Acknowledgements

Our research is supported by the PRISM (Protein Interactions and Stability in Medicine and Genomics) centre funded by the Novo Nordisk Foundation (NNF18OC0033950 to A.S. and K.L.L.), and by grants from the Novo Nordisk Foundation (NNF20OC0062606 and NNF18OC0052719 to W.B.) and the Lundbeck Foundation (R272-2017-4528 to A.S.).

## Additional information

### Funding

| Funder | Grant reference number | Author |
| --- | --- | --- |
| Novo Nordisk Fonden | NNF18OC0033950 | Amelie Stein<br>Kresten Lindorff-Larsen |
| Novo Nordisk Fonden | NNF20OC0062606 | Wouter Boomsma |
| Novo Nordisk Fonden | NNF18OC0052719 | Wouter Boomsma |
| Lundbeckfonden | R272-2017-4528 | Amelie Stein |

The funders had no role in study design, data collection and interpretation, or the decision to submit the work for publication.

### Author contributions

Lasse M Blaabjerg, Data curation, Software, Formal analysis, Investigation, Methodology, Writing - original draft, Writing – review and editing; Maher M Kassem, Software, Methodology, Writing – review and editing; Lydia L Good, Matteo Cagiada, Data curation, Formal analysis, Investigation, Writing – review and editing; Nicolas Jonsson, Formal analysis, Investigation, Visualization, Writing – review and editing; Kristoffer E Johansson, Resources, Data curation, Formal analysis, Writing – review and editing; Wouter Boomsma, Conceptualization, Resources, Software, Supervision, Funding acquisition, Methodology, Project administration, Writing – review and editing; Amelie Stein, Conceptualization, Supervision, Funding acquisition, Methodology, Project administration, Writing – review and editing; Kresten Lindorff-Larsen, Conceptualization, Resources, Data curation, Supervision, Funding acquisition, Methodology, Writing - original draft, Project administration, Writing – review and editing

### Author ORCIDs

Lasse M Blaabjerg (iD) http://orcid.org/0000-0003-0720-8632
Lydia L Good (iD) http://orcid.org/0000-0001-5308-8542
Nicolas Jonsson (iD) http://orcid.org/0000-0002-7838-1814
Kristoffer E Johansson (iD) http://orcid.org/0000-0001-6054-0461
Amelie Stein (iD) http://orcid.org/0000-0002-5862-1681
Kresten Lindorff-Larsen (iD) http://orcid.org/0000-0002-4750-6039

### Decision letter and Author response

Decision letter https://doi.org/10.7554/eLife.82593.sa1
Author response https://doi.org/10.7554/eLife.82593.sa2

## Additional files

### Supplementary files
• MDAR checklist

## Data availability

Scripts and data to repeat our analyses are available via: https://github.com/KULL-Centre/_2022_ ML-ddG-Blaabjerg/ (copy archived at *Linderstrøm-Lang Centre for Protein Science, University of Copenhagen, 2023*). Data is available at http://doi.org/10.17894/ucph.7f82bee1-b6ed-4660-8616-96482372e736. A browsable version is available at https://sid.erda.dk/sharelink/fFPJWflLeE.

The following dataset was generated:

| Author(s) | Year | Dataset title | Dataset URL | Database and Identifier |
|---|---|---|---|---|
| Blaabjerg LM, Kassem MM, Good LL, Jonsson N, Cagiada M, Johansson KE, Boomsma W, Stein A, Lindorff-Larsen K | 2022 | Supporting data for Blaabjerg et al. | http://doi.org/10.17894/ucph.7f82bee1-b6ed-4660-8616-96482372e736 | ERDA, 10.17894/ucph.7f82bee1-b6ed-4660-8616-96482372e736 |

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
