## [Editor Report]

This is a valuable study of broad potential impact in structural biology, which will interest readers in various fields, including molecular biomedicine, molecular evolution, and protein engineering.

---

## [Decision Letter]

**Decision letter after peer review:**

Thank you for submitting your article "Rapid protein stability prediction using deep learning representations" for consideration by *eLife*. Your article has been reviewed by 3 peer reviewers, and the evaluation has been overseen by José Faraldo-Gómez as the Senior Editor. Two of the reviewers agreed to reveal their identity, namely Nir Ben-Tal (Reviewer #1) and Julian Echave (Reviewer #2).

As you will see below, the reviewers conclude the manuscript is not suitable for publication for *eLife* in its current form, but they make specific recommendations to resolve their concerns. I recognize some of these revisions are substantial and will require considerable effort – nevertheless, I encourage you to take the time required to address these concerns convincingly, and then submit a revised version.

*Reviewer #1 (Recommendations for the authors):*

(1) Authors should compare the results to existing alternatives. For example, here is a recent method, also based on deep learning: https://journals.plos.org/ploscompbiol/article?id=10.1371/journal.pcbi.1008291. And there are many studies using 'simple' machine learning methods. The prediction quality here is similar to what these "shallow learning" methods give, albeit on other datasets, so perhaps irrelevant comparison. Anyway, all these methods show prediction quality that is very close to the natural upper bound given how noisy the data is (PMID: 30329016; a paper that I co-authored). So it is not clear whether the deep learning helped at all.

(2) Why was ROSETTA used as gold standard? Clearly ROSETTA is very far from perfect. In fact it's not even clear that it outperforms FoldX.

(3) Estimating ddG based on model structures: Now that AlphaFold models are available essentially to all proteins, it is possible to do much more than only 3 proteins. The authors should take advantage of this and examine the dependence of the ddG prediction quality on model RMSD to experimental structure, and more importantly, the dependence on AlphaFold's estimated model accuracy.

(4) Figure 4A. Both disease-causing and benign mutations appear to destabilize proteins. Authors claim that, on average, disease-causing mutations are more destabilizing. The calculated ddG values ~0.5 vs. ~1.4 kcal/mol are not that different. Is the difference really significant? More importantly, that the benign mutations appear to be destabilizing suggests that there might be systematic error towards destabilization. Could the authors test that? Do experimental ddG data show a similar behavior?

*Reviewer #2 (Recommendations for the authors):*

## Suggestions for improved or additional experiments data or analyses

1. More thorough assessment of accuracy against experiment, as compared not only with Rosetta, but also with other current methods. I think this work is much more likely to have an impact if a more thorough assessment of RaSP is made. In particular, I suggest that RaSP is more thoroughly tested against experimental mutational stability changes. In addition, to help position RaSP in the landscape of current methods, I suggest the RaSP-vs-experiment performance is compared not only to Rosetta, but also to other current methods. Such extra work does not necessarily mean that other methods need to be run. For instance, running RaSP on the same datasets used by recent assessments it would probably be possible to compare RaSP with other methods by getting the accuracy of other methods from the literature (see e.g. [https://doi.org/10.1093/bib/bbab555](https://doi.org/10.1093/bib/bbab555) for a recent thorough comparison of several methods).

2. Assess whether the method satisfies the antisymmetry condition. I think it would be important to complement Pearson's correlation and MAE with other measures of accuracy. Of particular importance is how well RaSP satisfies the thermodynamic condition that the reverse stability change satisfies ΔΔG(B->A) = – ΔΔG(A->B). Violation of this condition is a well-known issue with most ΔΔG-prediction methods (and experimental values!) and it would be very interesting to know whether RaSP suffers from this problem.

3. Compare the accuracy of the method for different sites. In the paper, there is an interesting analysis of the relative performance of RaSP for different types of amino-acid substitution. Another issue the authors might want to consider is how performance depends on whether sites are buried or exposed.

4. Compare the speed of the method to that of other methods. A strength of this work is that it presents a rapid method. From the present manuscript, it is clear that it is much more rapid than Rosetta. However, it is not clear how the speed of RaSP compares with other currently available methods. If the authors could perform such a comparison, at least for a limited dataset of proteins, I think this may go a long way towards making this work more influential.

## Recommendations for improving the writing and presentation

In general, I think the paper is very well written. However, some of what I perceived to be weaknesses could probably be dealt with by revising the manuscript.

When considering these recommendations, please take into account that my take on this work is that it is about a new rapid and accurate ΔΔG-prediction method. Therefore, I think everything in the manuscript should serve the purpose of bolstering this point. For instance, from this perspective, the calculations performed on the dataset of human proteins should be included only because it serves the purpose of demonstrating the usefulness of this approach, not because the obtained results are interesting in themselves. Therefore, I make a few recommendations that tend toward giving less weight to this application and more weight to the method and its assessment.

1. Make the application to the human dataset more clearly support the main point: the method. I think what's interesting about the large-scale application is not that common variants tend to be more stable, rare and/or disease variants unstable, etc, which we already know (you and others have demonstrated this already in previous work), but that it can be done by RaSP in a very short time because RaSP is fast. Also, the fact that it confirms previous findings shows that its accuracy is enough to draw these important conclusions. I think this is pretty clear from the paper. However, it can be made even more clear. For instance, almost half the abstract talks about the conclusions of the application, rather than the method itself. Also, it is not perfectly clear, from the Abstract, whether the method was developed as a mere tool to perform these calculations, or the calculations were performed to further assess and illustrate the method's usefulness. I think some minor carefully thought-out revisions in the abstract, results, and Discussion sections may make the method more of a protagonist and give the application, in the present paper, a supporting role.

2. Improve the method's description. From the perspective that the method is the point of the paper, I think the description may be much improved. Figure 1 may be much more detailed, for example. To understand the method I had to read the 2017 Paper by Boomsa and Frellsen, which is very nice. It would be great if some versions of Figure 3 and Figure 5 of that paper can be added to Figure 1. Also, I don't think it says anywhere in the present manuscript what sort of grid was used in the 3D-CNN part of the mode (which of the 3 grids used in the 2017 paper was used here?)

3. Highlight comparison with experiment. To support that the method is "accurate", comparison with experimental data is paramount, in my view. As I said before, I suggest you do a more thorough comparison. However, even if you decide not to, I suggest that you highlight the assessment that you have performed, by moving Table S1 and Figures S2 and S3 into the main document (If there are any limitations to the number of figures, I would rather get rid of Figure 4 than have Table S1 and Figures S2 and S3 as supplementary). Perhaps you could merge all data in Figures S2, Figures S3, and Table S1 into a single multi-panel figure that deals with all the comparison-with-experiment data.

4. Position the method in the landscape of current methods. As I said before, I think this work will be much more impactful if RaSP is positioned in the context of current methods. I suggested above that you do this by running some extra analyses and leveraging recent thorough method comparisons from the literature. If you decide not to do so, I think you could add a more detailed discussion of how you expect RaSP to compare with other methods regarding accuracy and computation speed.

5. Improve the Discussion. The Discussion, (and the Abstract) may be improved with little changes that make clearer that the point is the method, its accuracy, and speed and that the application is an example that supports that the method is useful, accurate, and fast. Also, depending on whether or not extra analyses are run, some discussion may be added to put this method in relation to other current methods.

*Reviewer #3 (Recommendations for the authors):*

[1] The main advantage of the method the authors claim is the speed. But they don't really take advantage of this. For example, I think the impact of this work would be much greater if they were to provide ddG predictions for all substitutions in all human proteins and to make these predictions available through a website and downloadable table. I don't see any reason not to do this, given it is not much work.

[2] The comparison to existing methods is rather anecdotal. How does the performance and run time compare to foldX? To MuteCompute? To other methods? How well do state of the art protein language models predict ddGS? How well do variant effect predictors such as CADD or Eve with good performance on the task of distinguishing pathological variants perform for predicting ddGs?

[3] How well does RASP perform for discriminating pathological from benign variants? The authors approach this problem but don't actually present the predictive performance nor do they compare it to other methods.

[4] The reality is that none of the methods for predicting ddGs perform particularly well and in places the text overstates the utility of computational methods. I would suggest the authors are more critical in their evaluation of the current state of the field. The real challenge – which is not addressed here – still remains, which is how to improve the predictive performance beyond the ok/reasonable but not that useful for many tasks predictions of Rosetta etc.

---

## [Author Response]

Reviewer #1 (Recommendations for the authors):(1) Authors should compare the results to existing alternatives. For example, here is a recent method, also based on deep learning: https://journals.plos.org/ploscompbiol/article?id=10.1371/journal.pcbi.1008291. And there are many studies using 'simple' machine learning methods. The prediction quality here is similar to what these "shallow learning" methods give, albeit on other datasets, so perhaps irrelevant comparison. Anyway, all these methods show prediction quality that is very close to the natural upper bound given how noisy the data is (PMID: 30329016; a paper that I co-authored). So it is not clear whether the deep learning helped at all.

We agree that the comparisons to other methods are important, but (as the reviewer knows well) also difficult to interpret due to issues outlined above. As suggested by reviewer 2, we have now used RaSP on a dataset that other methods have also been tested on [https://doi.org/10.1093/bib/bbab555]. This gives the reader the ability to evaluate RaSP in the context of other methods though we note (also in the revised manuscript) that differences in training/parametrization methods and data makes it difficult to compare. Indeed, we note also that while RaSP is not trained directly on experiments, it is also not fully independent because Rosetta is in itself based on experiments (including to some extent stability data).

(2) Why was ROSETTA used as gold standard? Clearly ROSETTA is very far from perfect. In fact it's not even clear that it outperforms FoldX.

A simple answer—as also outlined above—is that while Rosetta is indeed not perfect, it has worked well for us in a range of applications (as has FoldX). We suspect that using for example FoldX would have led to a similar model.

A slightly more complex explanation, which we now realize we did not present well in the original paper, is that the training on Rosetta is to some extent only a “refinement” and “re-scaling” of the structure-based convolutional neural network (CNN) that we use as input to the downstream neural network. This is important because the CNN is only trained on protein structures and has never seen experimental ∆∆G values. Indeed, a raw CNN similar to the one that we use achieves a relatively good correlation with experiments [https://dl.acm.org/doi/10.5555/3294996.3295102], as also seen before training of the model against Rosetta data [Figure 2 of our paper]. Training of the downstream neural network thus acts to fine-tune the information contained in the CNN and to bring predictions onto a common, thermodynamic scale.

We now explain these points in the revised paper.

(3) Estimating ddG based on model structures: Now that AlphaFold models are available essentially to all proteins, it is possible to do much more than only 3 proteins. The authors should take advantage of this and examine the dependence of the ddG prediction quality on model RMSD to experimental structure, and more importantly, the dependence on AlphaFold's estimated model accuracy.

Before answering this point in more detail, we first note (i) that we have recently tested the accuracy on Rosetta (and other stability prediction methods) on model quality using experimental data [https://doi.org/10.1101/2022.07.12.499700] with an analysis similar to that in Figure 3 of the paper and (ii) that stability calculations using Rosetta (and FoldX) using high-quality AlphaFold2 structures as input give as accurate results as using experimental structures [https://doi.org/10.1101/2021.09.26.461876]. Given these results (which suggest that high-quality predicted structures would work well as input to RaSP)—as well as the greater computational costs of running Rosetta on a large number of proteins—we initially decided only to validate on a smaller number of proteins.

Based on the reviewer’s comment, we have however now expanded this analysis. We note, however, that because estimated model accuracy (e.g. pLDDT) is correlated also to protein structure (e.g. disordered or mobile regions tend to have low pLDDT), it is not trivial to assess to what extent prediction accuracy depends on pLDDT and why. That is, low-pLDDT regions will often have small ∆∆G values, which might mean that the MSE would be small but correlations weak, simply because the dynamical range is different.

We have thus expanded the analyses we performed to six proteins, selecting again proteins for which at least two crystal structures exist to set a reasonable baseline to what a good prediction would look like. We now also break down the analysis depending on AlphaFold 2’s predicted accuracy (pLDDT score). The results show that ∆∆G predictions are more consistent (i.e. better correlated) in regions of the AlphaFold 2 structures that have high pLDDT compared to regions that have low pLDDT.

(4) Figure 4A. Both disease-causing and benign mutations appear to destabilize proteins. Authors claim that, on average, disease-causing mutations are more destabilizing. The calculated ddG values ~0.5 vs. ~1.4 kcal/mol are not that different. Is the difference really significant? More importantly, that the benign mutations appear to be destabilizing suggests that there might be systematic error towards destabilization. Could the authors test that? Do experimental ddG data show a similar behavior?

While the difference in the median is indeed only ca. 1 kcal/mol, this is highly statistically significant. For example, using bootstrapping we find that a lower bound of the difference in medians is 0.73 using a 95% confidence interval. These results have now been added to paper.

Concerning the point that many benign variants are (mildly) destabilizing, we note that mild (predicted) destabilization tends not to cause substantial loss of function in itself as assed both by multiplexed assays of variant effects

[https://doi.org/10.1016/j.celrep.2021.110207] or via analysis of pathogenicity in several proteins [https://doi.org/10.1371/journal.pgen.1006739 and https://doi.org/10.7554/*eLife*.49138 are two examples from our own research]. Thus, these results are not different from those using other stability predictors.

Regarding the point of whether there is a bias relative to experiments, this is difficult to answer because experimental data is sparse and noisy; specifically we do not know of any systematic analysis of measured ∆∆G values for a large range of pathogenic and benign variants. To examine this question, we instead calculated both the (signed) mean error and (unsigned) mean absolute error between experimental and calculated values for both Rosetta and RaSP using the data presented in Figure S2. The results show that RaSP does not appear to be biased relative to for example Rosetta.

Reviewer #2 (Recommendations for the authors):## Suggestions for improved or additional experiments data or analyses1. More thorough assessment of accuracy against experiment, as compared not only with Rosetta, but also with other current methods. I think this work is much more likely to have an impact if a more thorough assessment of RaSP is made. In particular, I suggest that RaSP is more thoroughly tested against experimental mutational stability changes. In addition, to help position RaSP in the landscape of current methods, I suggest the RaSP-vs-experiment performance is compared not only to Rosetta, but also to other current methods. Such extra work does not necessarily mean that other methods need to be run. For instance, running RaSP on the same datasets used by recent assessments it would probably be possible to compare RaSP with other methods by getting the accuracy of other methods from the literature (see e.g. [https://doi.org/10.1093/bib/bbab555](https://doi.org/10.1093/bib/bbab555) for a recent thorough comparison of several methods).

We thank the reviewer for this suggestion. We have now analysed the S669 dataset [from https://doi.org/10.1093/bib/bbab555] using RaSP, and report the results in the revised manuscript. These results enable us to compare RaSP to other methods.

2. Assess whether the method satisfies the antisymmetry condition. I think it would be important to complement Pearson's correlation and MAE with other measures of accuracy. Of particular importance is how well RaSP satisfies the thermodynamic condition that the reverse stability change satisfies ΔΔG(B->A) = – ΔΔG(A->B). Violation of this condition is a well-known issue with most ΔΔG-prediction methods (and experimental values!) and it would be very interesting to know whether RaSP suffers from this problem.

We have now also analysed RaSP on pairs of structures that differ by a single substitution including also the Ssym+ data set [from https://doi.org/10.1093/bib/bbab555], again noting the bias in that most original variants are deletion variants.

3. Compare the accuracy of the method for different sites. In the paper, there is an interesting analysis of the relative performance of RaSP for different types of amino-acid substitution. Another issue the authors might want to consider is how performance depends on whether sites are buried or exposed.

We have added an analysis on the accuracy (as assessed by agreement with Rosetta) depending on whether the sites are buried or exposed. Interpretation of the results is complicated by the fact that the ∆∆G range is very different in these two parts of proteins, and so we report both correlation coefficients and MAE values. We find that RaSP correlations with Rosetta are generally lower at buried residues compared those exposed on the surface – though this may also be due to the higher variance of the target Rosetta ∆∆G values in those regions.

4. Compare the speed of the method to that of other methods. A strength of this work is that it presents a rapid method. From the present manuscript, it is clear that it is much more rapid than Rosetta. However, it is not clear how the speed of RaSP compares with other currently available methods. If the authors could perform such a comparison, at least for a limited dataset of proteins, I think this may go a long way towards making this work more influential.

While we agree that this is an interesting question, we think that assessing the speed of a large number of other methods is outside the scope of our work. Such analyses are also complicated by the fact that many other methods are only available as webservices. Also, speed comparisons will depend on the chosen hardware since, for example, not all methods can exploit the parallel nature of GPUs. For these reasons, we have opted to examine three additional methods, namely the widely used FoldX, [https://doi.org/10.1093/nar/gki387] as well as two neural-network based methods ACDC-NN [https://doi.org/10.1088/13616463/abedfb] and ThermoNet [https://doi.org/10.1371/journal.pcbi.1008291]. The results are shown in the revised paper.

## Recommendations for improving the writing and presentationIn general, I think the paper is very well written. However, some of what I perceived to be weaknesses could probably be dealt with by revising the manuscript.When considering these recommendations, please take into account that my take on this work is that it is about a new rapid and accurate ΔΔG-prediction method. Therefore, I think everything in the manuscript should serve the purpose of bolstering this point. For instance, from this perspective, the calculations performed on the dataset of human proteins should be included only because it serves the purpose of demonstrating the usefulness of this approach, not because the obtained results are interesting in themselves. Therefore, I make a few recommendations that tend toward giving less weight to this application and more weight to the method and its assessment.1. Make the application to the human dataset more clearly support the main point: the method. I think what's interesting about the large-scale application is not that common variants tend to be more stable, rare and/or disease variants unstable, etc, which we already know (you and others have demonstrated this already in previous work), but that it can be done by RaSP in a very short time because RaSP is fast. Also, the fact that it confirms previous findings shows that its accuracy is enough to draw these important conclusions. I think this is pretty clear from the paper. However, it can be made even more clear. For instance, almost half the abstract talks about the conclusions of the application, rather than the method itself. Also, it is not perfectly clear, from the Abstract, whether the method was developed as a mere tool to perform these calculations, or the calculations were performed to further assess and illustrate the method's usefulness. I think some minor carefully thought-out revisions in the abstract, results, and Discussion sections may make the method more of a protagonist and give the application, in the present paper, a supporting role.

We agree that the paper is mostly about the method, realizing also that showing why we developed the method is an important point. We have attempted to update the manuscript with the suggestions by the reviewer in mind, and hopefully put the method itself in greater focus.

2. Improve the method's description. From the perspective that the method is the point of the paper, I think the description may be much improved. Figure 1 may be much more detailed, for example. To understand the method I had to read the 2017 Paper by Boomsa and Frellsen, which is very nice. It would be great if some versions of Figure 3 and Figure 5 of that paper can be added to Figure 1. Also, I don't think it says anywhere in the present manuscript what sort of grid was used in the 3D-CNN part of the mode (which of the 3 grids used in the 2017 paper was used here?)

We have expanded the description of the method, and also updated Figure 1 to give a better schematic description of the 3D CNN, while also keeping the context for the full model including training against Rosetta. The type of grid is now also explicitly specified.

3. Highlight comparison with experiment. To support that the method is "accurate", comparison with experimental data is paramount, in my view. As I said before, I suggest you do a more thorough comparison. However, even if you decide not to, I suggest that you highlight the assessment that you have performed, by moving Table S1 and Figures S2 and S3 into the main document (If there are any limitations to the number of figures, I would rather get rid of Figure 4 than have Table S1 and Figures S2 and S3 as supplementary). Perhaps you could merge all data in Figures S2, Figures S3, and Table S1 into a single multi-panel figure that deals with all the comparison-with-experiment data.

As described above, we have added a more extensive comparisons to experiments. The results previously shown in Figures S2 are now shown in the main text.

4. Position the method in the landscape of current methods. As I said before, I think this work will be much more impactful if RaSP is positioned in the context of current methods. I suggested above that you do this by running some extra analyses and leveraging recent thorough method comparisons from the literature. If you decide not to do so, I think you could add a more detailed discussion of how you expect RaSP to compare with other methods regarding accuracy and computation speed.

Using the results on S669 dataset as well as the speed assessments for additional methods, we now provide an expanded discussion of how we think RaSP can be used.

5. Improve the Discussion. The Discussion, (and the Abstract) may be improved with little changes that make clearer that the point is the method, its accuracy, and speed and that the application is an example that supports that the method is useful, accurate, and fast. Also, depending on whether or not extra analyses are run, some discussion may be added to put this method in relation to other current methods.

As discussed above, we have focused the paper more on the method and validation.

Reviewer #3 (Recommendations for the authors):[1] The main advantage of the method the authors claim is the speed. But they don't really take advantage of this. For example, I think the impact of this work would be much greater if they were to provide ddG predictions for all substitutions in all human proteins and to make these predictions available through a website and downloadable table. I don't see any reason not to do this, given it is not much work.

We have now applied RaSP to all human protein structures as predicted using AlphaFold 2, and now make these available for download and show an overall histogram in the manuscript. We note, however, that we do not expect RaSP’s accuracy to uniform across all of these proteins; for example RaSP hasn’t been trained to predict stability changes in membrane proteins. Also, ∆∆G predictions for disordered proteins (in their free state) are poorly defined and AlphaFold is not expected to give reasonable models for such disordered regions (or other regions with low confidence scores). We have thus added caveats to the uses of this data in the text.

[2] The comparison to existing methods is rather anecdotal. How does the performance and run time compare to foldX? To MuteCompute? To other methods? How well do state of the art protein language models predict ddGS? How well do variant effect predictors such as CADD or Eve with good performance on the task of distinguishing pathological variants perform for predicting ddGs?

There are two slightly different points here. First, regarding how well RaSP compares to predict stability changes relative to other methods, we have (as described in our response to reviewer 2) run RaSP on a larger validation set, which enables comparison with other methods. We have also performed comparisons to run times of other methods. As for the question on using non-stability predictors to predict stability changes, we have not analysed this. While these methods might predict stability changes, their scores and goal are very different, and we do not find compelling reasons to attempt to apply them to this problem.

[3] How well does RASP perform for discriminating pathological from benign variants? The authors approach this problem but don't actually present the predictive performance nor do they compare it to other methods.

The goal of our ∆∆G predictions for pathogenic/benign variants was not to assess how well RaSP performs as a variant effect predictor on its own. Indeed, we and others have previously shown (on a much smaller set of proteins) that these methods work relatively well, but not quite as well as methods that are for example based on analysis of multiple sequence alignments. Indeed, while loss of abundance due to loss of stability is one very common mechanism for pathogenicity, there are many others (loss of intrinsic function or binding to other proteins); only the first category is captured by RaSP, whereas evolutionary-based methods and various meta-predictors capture all of these mechanisms. However, as we have previously shown, combining stability predictions with evolutionary assessments enables both accurate predictions and assessment of mechanism. Thus, RaSP is an important step in finding those variants that cause disease via loss of abundance. We now explain these issues better in the revised manuscript, but as suggested by Reviewer 2 we also focus less on the application and more on the method.

[4] The reality is that none of the methods for predicting ddGs perform particularly well and in places the text overstates the utility of computational methods. I would suggest the authors are more critical in their evaluation of the current state of the field. The real challenge – which is not addressed here – still remains, which is how to improve the predictive performance beyond the ok/reasonable but not that useful for many tasks predictions of Rosetta etc.

We agree that none of the available methods work as well as one might hope, though also note (in the revised manuscript and in the responses above) that limitations in the data also limits how well we can assess the methods.